

# A novel adaptive weight bi-directional long short-term memory (AWBi-LSTM) classifier model for heart stroke risk level prediction in IoT

S Thumilvannan[1] and R Balamanigandan[2]

[1] Department of Computer Science and Engineering, Saveetha School of Engineering, Saveetha Institute of Medical and Technical Sciences, SIMATS, Saveetha University, Chennai, Tamilnadu, India
[2] Department of Computer Science and Engineering, Saveetha School of Engineering, Saveetha Institute of Medical and Technical Sciences, SIMATS, Saveetha University, Chennai, Tamilnadu, India

Corresponding author
R Balamanigandan,
balamanigandanr.sse@saveetha.com

## ABSTRACT

Stroke prediction has become one of the significant research areas due to the increasing fatality rate. Hence, this article proposes a novel Adaptive Weight Bi-Directional Long Short-Term Memory (AWBi-LSTM) classifier model for stroke risk level prediction for IoT data. To efficiently train the classifier, Hybrid Genetic removes the missing data with Kmeans Algorithm (HKGA), and the data are aggregated. Then, the features are reduced with independent component analysis (ICA) to reduce the dataset size. After the correlated features are identified using the T-test-based uniform distribution-gradient search rule-based elephant herding optimization for cluster analysis (GSRBEHO) (T-test-UD-GSRBEHO). Next, the fuzzy rule-based decisions are created with the T-test-UDEHOA correlated features to classify the risk levels accurately. The feature values obtained from the fuzzy logic are given to the AWBi-LSTM classifier, which predicts and classifies the risk level of heart disease and diabetes. After the risk level is predicted, the data is securely stored in the database. Here, the MD5-Elliptic Curve Cryptography (MD5-ECC) technique is utilized for secure storage. Testing the suggested risk prediction model on the Stroke prediction dataset reveals potential efficacy. By obtaining an accuracy of 99.6%, the research outcomes demonstrated that the proposed model outperforms the existing techniques.

## INTRODUCTION

The Internet of Things (IoT) has been widely used in several applications, particularly in recent decades (*Gubbi et al., 2013*). Many medical applications now extensively utilize IoT to support individuals with their health conditions. As a result, the importance of mobile health services has increased, as they are crucial for monitoring and managing patients with chronic conditions such as diabetes and cardiovascular disease (*Yuehong et al., 2016*; *Guariguata et al., 2014*). In smart health, specifically in patient monitoring, it is essential that patient data are handled properly. Big data techniques enable patients

with specific diseases to receive preventative medications—for example, for heart failure, which can be caused by diabetes or hypertension (*Dhillon & Kalra, 2017*). There are many chronic diseases such as diabetes, cancer, heart disease, and stroke that should receive more attention due to their high fatality rates. Chronic disease is a deadly illness that has recently topped the global list of killers and requires intensely vigilant surveillance to keep patients healthy (*Rghioui et al., 2019*).

A significant risk factor for stroke is diabetes mellitus, characterized by chronic hyperglycemia caused by an absolute or relative insulin deficit. There is a two to five-fold increased risk of stroke in those with diabetes compared to those without the disease. Cardiovascular risk reduction measures focus on preventing the development, recurrence, and progression of acute stroke through extensive clinical trials conducted in adults with diabetes.

According to *Benjamin et al. (2017)*, a new or recurrent stroke affects 795,000 individuals annually in the US, with one case occurring every 40 s on average. In the first year after a stroke, one out of every five victims dies (*Koton et al., 2014*). The burden of paying for the survivors' rehabilitation and health care falls heavily on their families and the medical field. From 2014 to 2015, stroke-related direct and indirect expenditures amounted to approximately 45.5 billion US dollars (*Benjamin et al., 2019*). Accurate stroke prediction is essential to reduce the expense of early medications and to minimize the risks of stroke. Electronic health records and retinal scans are just two examples of the medical data used to construct Stroke Risk Prediction (SRP) algorithms. Deep learning and conventional machine learning techniques such as support vector machine (SVM), decision tree, and logistic regression are widely used in healthcare applications (*Khosla et al., 2010*; *Monteiro et al., 2018*; *Sung, Lin & Hu, 2020*). The best results for stroke prediction have reportedly been attained by deep neural networks (DNN) (*Cheon, Kim & Lim, 2019*). However, it can be challenging to find the volume of reliable data required in a practical situation (*Wang, Casalino & Khullar, 2019*). The strict privacy protection laws in the medical field make it difficult for hospitals to share stroke data.

Small subsets of the complete database of stroke data are usually scattered among numerous institutions. Moreover, stroke statistics may show extremely imbalanced positive and negative cases.

Machine learning (ML) techniques are typically selected for enhancing patient care because they deliver faster, more accurate results. Due to its distinctive capacity to integrate data from numerous sources and handle vast amounts of data, deep learning (DL) improves predictive features (*Nasser et al., 2021*). However, they take longer to learn and evaluate data, have long prediction periods, and use many processing resources for training and recognition. Age, smoking, hypertension, cholesterol, and diabetes are among the established risk variables that were forecasted in earlier models for forecasting the chance of acquiring diabetes and heart disease. To determine if those with both risk of cardiovascular disease and isolated impaired fasting or impaired glucose tolerance, they did not include those with both as a separate group for analysis (*Kumar et al., 2021*).

Hence, a novel framework named an Adaptive Weight Bi-Directional Long Short-Term Memory (AWBi-LSTM) classifier-based stroke risk level prediction model for IoT data is

proposed in this article. Here, to efficiently train the classifier, the HGKA algorithm removes the missing data, and the data are aggregated. After that, the characteristics are minimized using independent component analysis (ICA). After the correlated features are identified using the T-test-based Uniform Distribution-gradient search rule-based Elephant Herding Optimization for cluster analysis (T-test-UD-GSRBEHO). The AWBi-LSTM classifier is used to predict and classify the risk level of diabetes and heart disease based on the feature values derived from the fuzzy logic. Data is safely saved in the database after the risk level has been estimated. Here, more accurate safe storage is achieved by the use of the MD5-Elliptic Curve Cryptography (MD5-ECC) technology.

The main contribution of the article are as follows:

- Predicting strokes and analyzing medical data is significant as it enables the early identification of individuals at high risk, allowing for rapid treatment and preventive measures.
- The AWBi-LSTM technique is deployed to improve the accuracy of risk assessments for illnesses, taking into account various risk factors among local populations.
- The proposed model uses a dataset with attributes such as patient's gender, age, medical history, and smoking status to determine the likelihood of a stroke. Important patient information is included in the data packet.

The article is structured in the following manner: 'Literature Review' analyses the pros and cons of the existing works associated with the proposed method. 'Proposed Methodology' clearly describes methodology of the proposed model. 'Results and Discussion' analyses the efficiency of the proposed model through simulation results. Finally, 'Conclusion' concludes the research with the findings.

## LITERATURE REVIEW

Data mining algorithms can forecast heart disease and diabetes in patients based on patient medical information. The most recent findings on heart disease prediction utilizing deep learning and machine learning methods are reviewed in this section. *Hossen et al. (2021)* conducted a survey divided into three categories: deep learning models for CVD prediction, machine learning models for CVD, and classification and data mining methodologies. Additionally, this research gathers and summarizes the tools used for each group of approaches as well as the datasets used for classification and prediction, and the outcome metrics for reporting accuracy.

*Ahuja, Sharma & Ali (2019)* utilized SVM, MLP, random forest, logistic regression, and decision tree, among other techniques. The PIMA dataset was used to forecast patients' diabetes more precisely. Significant results for naive Bayes were obtained by another investigation that employed the PIMA dataset (*Pranto et al., 2020*). The stacking method was used by *Kuchi et al. (2019)* to achieve a 95.4% accuracy. The diagnosis of diabetes requires more research, as stated by *Kavakiotis et al. (2017)*. By combining several classifiers, the precision of predicting diabetes may be enhanced. An accurate disease diagnosis is an essential component of medical care. Numerous researchers have produced inaccurate diagnostic tools for cardiac disease.

*Khan (2020)* developed a framework based on the IoT to address the accuracy issue associated with cardiac disease. This structure evaluates multiple risk factors for heart attacks, including blood pressure and electrocardiogram, using a heart monitor and smartwatch. Additionally, a more effective Deep CNN is used to accurately forecast heart attacks utilizing collected data. The IoT framework has an accuracy rate of over 95%.

*Pan et al. (2020)* introduced an improved deep learning and CNN-based method for successfully treating heart disease *via* the Internet of Things. The aforementioned combination aims to raise heart disease prognosis rates. The model's efficiency is calculated utilizing every disease-related attribute and its depreciation. Additionally, the suggested combination is implemented *via* IoMT, outcomes are assessed utilizing accuracy and processing speed, and the model yields improved outcomes.

*Ahmed et al. (2020)* demonstrated a real-time method for forecasting cardiac disease using data streams that included patients' current medical condition. The secondary objective of the study is to determine the most effective machine learning (ML) methods for heart disease prediction. To increase accuracy, ML algorithm parameters are also adjusted. According to the findings, the random forest has a greater accuracy rate than other ML techniques.

*Yu et al. (2020a)* and *Yu et al. (2020b)* created a stroke prediction method using each person's biosignals. Most stroke detection techniques consider visual data rather than biosignals. In addition, the prediction system incorporates deep learning and random forest algorithms for selecting the best features and performing the prediction task accordingly. Findings showed that the LSTM system obtained 93.8% accuracy, whereas the random forest-based system achieved 90.4% accuracy.

*Bhattacharya et al. (2020)* built a model using antlions and DNNs to manage the multimodality in the stroke dataset. This framework considers the antlion technique to optimize the hyperparameter DNN. Additionally, the parameter-tuned DNN is used to forecast data from strokes. When outcomes are compared to training time, it is found that the training time for that model is 38.13.

*Ali et al. (2020)* proposed a novel medical device to predict the probability of a heart attack. Combined with ensemble deep learning methods, this architecture supports feature fusion. The feature fusion approach involves fusing attribute information from electronic records and sensor data. Additionally, the data-gathering strategy eliminates irrelevant data. The algorithm is further developed *via* ensemble deep learning for even better outcomes. Simulation findings demonstrate the value of an intelligent medical system for forecasting heart attacks.

Heart disease and stroke are the second leading causes of death (*Moghadas, Rezazadeh & Farahbakhsh, 2020*). If the condition is not identified in time, it worsens. Therefore, an IoT and fog-based system for accurate diagnosis was created, taking the detection rate of heart disease into consideration as a potential problem. Additionally, ECG signals are considered for the accurate and prompt detection of cardiac illness, and k-NN is used to validate the previously described framework.

*Yu et al. (2020a)* and *Yu et al. (2020b)* demonstrated the effects of stroke severity on elderly persons older than 65 using the NIHSS. The C4.5 algorithm is taken into

consideration to determine how severe a stroke will be for elderly people. In addition, thirteen rather than the eighteen elements of the stroke scale are included in the assessment, which shows that C4.5 has a 91.11% accuracy rate.

*Yahyaie, Tarokh & Mahmoodyar (2019)* examined the effectiveness of an IoT model for accurately predicting cardiac illness. The ECG signal is considered in this research while assessing the model's efficacy. Utilizing a cloud-based internet application, a total of 271 people's data are gathered. Ninety features for heart disease are included in the collected dataset. Additionally, the IoT model is trained to utilize an NN approach, and it is stated that it achieves an acceptable accuracy level. Smart health products, IoT, IoMT, and intelligent ML approaches like ANN, DNN, CNN, etc., can greatly enhance healthcare systems.

According to the observations from the above-mentioned discussions, existing approaches have various drawbacks, such as lower accuracy and maximal time consumption. To overcome these constraints, a novel deep learning-based technique is presented to enhance heart disease prediction performance.

## PROPOSED METHODOLOGY

This article proposes a novel Adaptive Weight Bi-Directional Long Short-Term Memory (AWBi-LSTM) classifier-based stroke prediction model for IoT data. The proposed flow diagram is depicted in Fig. 1.

### Input stroke prediction dataset

Using the stroke prediction dataset, the efficacy of the suggested risk prediction method is evaluated. This dataset determines the likelihood of a person suffering from a stroke based on 11 input characteristics, including age, gender, profession, marital status, BMI, hypertension, glucose levels, chest discomfort, blood pressure, existing diseases, and smoking status. The dataset comprises more than 5,000 samples. The Kaggle Stroke Prediction dataset can be found here (https://www.kaggle.com/datasets/fedesoriano/stroke-prediction-dataset).

### Data-preprocessing

Initially, the input data in the dataset $I$ ispreprocessed to enhance the working efficacy of the classifier. In the proposed technique, preprocessing is done by removing the missing values and aggregating the data.

#### *Missing data removal using hybrid genetic with kmeans algorithm*

One of the popular clustering techniques is the K-means approach, which has been applied in various scientific and technological domains. The initial center vectors might cause empty clusters using the k-means algorithm. GAs are flexible heuristic search algorithms based on natural selection and genetics. The empty cluster problem is effectively solved by the hybrid k-means technique presented in this research, which is also used to cluster the data objects.

The following are the main issues with the K-means algorithm:

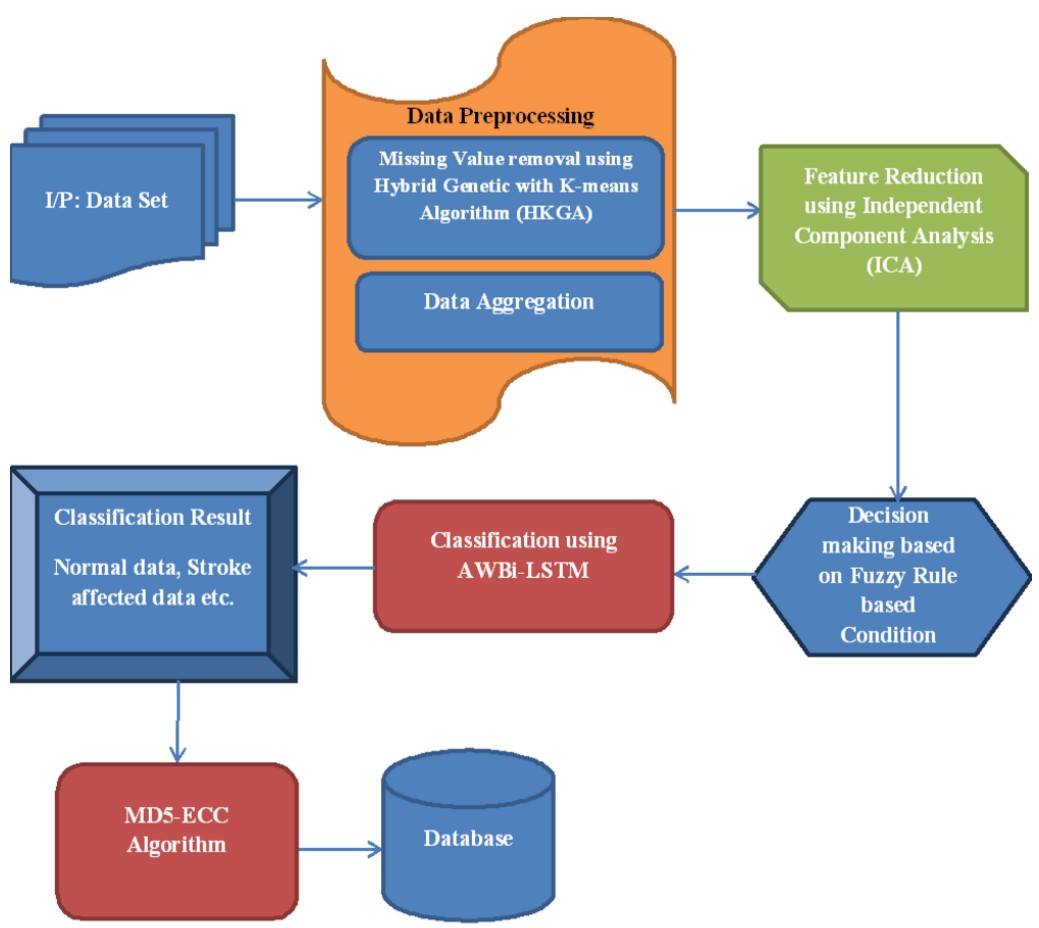

**Figure 1** Proposed flow diagram.

- Based on the original center vectors, it might yield empty clusters.
- It could converge to non-optimal values.
- With a considerable amount of computation work, finding global solutions to huge problems is impossible.

This work introduces the hybrid genetic algorithm (HKGA) to effectively address the above-mentioned disadvantages.

### Phase 1: K-means algorithm

Step 1: K initial cluster centres $z_1, z_2, z_3, \ldots, z_k$ are selected arbitrarily from the n observations $\{x_1, x_2, x_3, \ldots, x_n\}$.

Step 2: A point $x_1, \mathrm{i} = 1, 2, 3, \ldots, n$ is allotted to cluster $C_j, j \in \{1, 2, \mathrm{alk}\}$ if

$$\left\| x_i - z_j \right\| < \left\| x_i - z_p \right\|, p = 1, 2, \ldots, K \& j \neq p \tag{1}$$

Step 3: New cluster centres $z_1, z_2, z_3, \ldots, z_k$ are computed as follows:

$$z_j^* = \frac{1}{n_i} \sum_{x_i \in C_j} x_i, i = 1, 2, \ldots, K \tag{2}$$

where $n_i$ is the number of aspects that belong to cluster $C_j$.

Step 4: If $z_i^* = z_i, i = 1, 2, \ldots, K$ then stop; if not, move on to step 2.

obtain an initial center for each selected cluster following this procedure.

## Phase 2: genetic algorithm

Step 1: Population initialization

Every individual represents a row-matrix of 1$ven$ where $n$ is the number of observations, and each gene contains the integer $[1, K]$ that denotes the cluster to which this observation pertains. For example, suppose there are ten observations $\{x_1, x_2, \ldots, x_{10}\}$ that need to be allocated to four clusters $k = 4$.

Step 2: Evaluation

Find cluster classifications that minimize the fitness function based on the specified objective function. The fitness function for clustering in the K clusters Given is $C_1, C_2, C_3, \ldots, C_k$.

$$f(C_1, C_2, \ldots, C_k) = \sum_{i=1}^{k} \sum_{x_j \in C_i} \left\| x_j - z_i \right\| \tag{3}$$

Step 3: Selection

The purpose of selection is to focus GA search on interesting areas of the search field. In this work, roulette wheel selection is used, where individuals from each generation are chosen based on a probability value to survive into the following generation. According to the following formula, the likelihood of variable selection relates to the population's fitness value:

$$p(x) = \frac{f(x) - f_{Min}(\Psi)}{\sum_{x \in \Psi} \{f(x) - f_{Min}(\Psi)\}} \tag{4}$$

Where $p(x)$, string's selection probability $x$ in a population $\Psi$ and

$$f_{Min}(\Psi) = Min\{f(x) | x(x)\} \tag{5}$$

Step 4: Crossover operator

The crossover is performed on each individual in this stage using a modified uniform crossover, whereby the individual chosen with a probability is used to make the offspring.

Step 5: Mutation operator

An implementation of the mutation operator is utilized for each individual. First, choose two columns at random from the ith person. Then, generate two new columns.

Step 6: The most optimal solutions found so far throughout the procedure, as opposed to GA keeping the best solutions found among the current population.

*Data aggregation*

Once the missing data has been eliminated, the values are combined by applying mean ($\alpha$), median ($\beta$), variance ($\nu$) and standard deviation ($sd$), to establish the dataset.

$$\alpha = \frac{\sum_{x=1}^{b} m_x}{b} \tag{6}$$

$$\beta = median(m_x) \tag{7}$$

$$sd = \sqrt{\frac{\sum_{x=1}^{b} (m_x - \alpha)^2}{b}} \tag{8}$$

$$\nu^2 = \frac{\sum_{x=1}^{b} (m_x - \alpha)^2}{b-1}. \tag{9}$$

Thus, the preprocessed dataset ($Y$) is given as,

$$Y = \{K_1, K_2, \ldots, K_B\} \, or \, K_\nu, \nu = 1, 2, \ldots, B \tag{10}$$

where, $K_B$ represents the preprocessed $B^{th}$ patient data.

## Independent component analysis for feature reduction

ICA is an unsupervised feature extraction technique that has been applied to many applications. It transforms the original data using a transformation function. The model of the ICA is defined as,

$$Y = sX \tag{11}$$

where, Y–Transformed data. s–Scalar matrix. X–Original data.

Here, the original data is transformed into transformation data using the Tanh transformation function as a scalar function. The non-linearity among the data will be maximized, and orthogonality for each data vector will be achieved using this Tanh transformation function. Selecting the number of independent components is a critical problem in ICA. The components with an average greater than 0.1 in the newly transformed dataset are selected.

## Feature correlation using T-test-UD-GSRBEHO

After the feature reduction process, correlated features are identified using the T-test-based Uniform Distribution Gradient Search Rule-based Elephant Herding Optimization for Cluster Analysis (T-test-UD-GSRBEHO).

Initially, the obtained features $\{t_r\}$ undergo a $T$-test, and the $T$-test process is given as,

$$\tau_r = \frac{\bar{t}_r - \bar{t}_{r+1}}{\sqrt{\delta^2 ((d_r + d_{r+1})/d_r \times d_{r+1})}} \tag{12}$$

where, $\tau$ is the T-value for the feature $r,\delta$ depicts the pooled standard errors of $t_r, t_{r+1}$, and $d_r, d_{r+1}$ illustrates the overall quantity of data associated with the given attributes $t_r, t_{r+1}.\bar{t}_r, \bar{t}_{r+1}$ depicts the mean values of the features $t_r, t_{r+1}$ (*Thumilvannan & Balamanigandan, 2023*).

After the $\tau$ iscalculated for all samples, the Spearman correlation coefficient is used to assess the correlation between features.

$$\lambda_r = 1 - \frac{6\sum \tau_r^2}{l\left(l^2 - 1\right)} \tag{13}$$

where $l$ represents the overall quantity of characteristics. The non-zero values are mixed with the reduced characteristics among the linked features. This process yields a valid feature set. From the resulting feature sets, the optimal feature set is selected using the Uniform Distribution-Gradient Search Rule-Based Elephant Herding Optimization for cluster analysis (UD-GSRBEHO), as follows *Thumilvannan & Balamanigandan (2023)*.

**Initialization:** The feature sets obtained consist of an initial group of elephants with a predetermined number, denoted as,

$$U = \{[u_1], [u_2], \ldots\ldots, [u_d]\}\ or\ \left[u_\phi\right], \phi = 1, 2, \ldots, d \tag{14}$$

where, $U$ depicts the elephant population and $\left[u_\phi\right]$ depicts the $\phi^{th}$ elephant clan with comparable elephant numbers. Every generation of elephants sees the males departing from the tribe and relocating far from it, while the females stay with their group. The matriarch of each elephant tribe is in charge (*Thumilvannan & Balamanigandan, 2023*).

**Clan updating operator:** Every elephant in a clan has an estimated level of fitness. As the other elephants in the clan adjust their positions in accordance with the matriarch, the elephant deemed most fit is regarded as the matriarch (*Thumilvannan & Balamanigandan, 2023*). In this clan, fitness is the new status $\left[u_\phi\right]$ is given as $N_{\left[u_\phi\right],\omega}^{R+1}$ which is evaluated as,

$$N_{\left[u_\phi\right],\omega}^{R+1} = N_{\left[u_\phi\right],\omega}^{R} + \Omega\left(N_{\left[u_\phi\right]}^{*} - N_{\left[u_\phi\right],\omega}^{R}\right).\gamma. \tag{15}$$

Here, $R$ signifies the iteration, $N_{\left[u_\phi\right]}^{*}$ depicts the best solution of clan, $\Omega$ signifies the algorithm parameter, the presence of a matriarch in the group is $\gamma$ indicated, and the random number derived from a uniform distribution is denoted as,

$$\gamma = \frac{1}{N_{\left[u_\phi\right]+1}^{R} - N_{\left[u_\phi\right]}^{R}}. \tag{16}$$

The position of the best solution in each clan is updated with respect to the following equation,

$$N_{\left[u_\phi\right]}^{R+1} = \Delta.N_{\left[u_\phi\right]}^{c} \tag{17}$$

where, $\Delta$ the second algorithm parameter represents the degree of influence exerted by the clan center $N_{\left[u_\phi\right]}^{c}$. The clan center is mathematically represented as,

$$N_{\left[u_\phi\right],\Re}^{c} = \frac{1}{\eta_{\left[u_\phi\right]}}.\sum_{\omega=1}^{\eta_{\left[u_\phi\right]}} N_{\left[u_\phi\right],\omega,\Re} \tag{18}$$

where, $\Re$ represents the dimension of $N_{[u_\phi],\omega}$, and $\eta_{[u_\phi]}$ indicates elephants in the clan as a whole $[u_\phi]$.

**Clan separating operator:** Male elephants disassociate from the clan, a phenomenon that may be represented by the separation operator. In each cycle, the separation involves the elimination of the most problematic elephants from the clans,

$$N'_{[u_\phi]} = N_{mnm} + (N_{mxm} - N_{mnm} + 1) \tag{19}$$

where, $N_{mxm}, N_{mnm}$ depicts the upper and lower bound of the elephant in the clan $[u_\phi].N'_{[u_\phi]}$ indicates the worst elephant position in the clan $[u_\phi]$ which gets removed from the clan.

## Gradient search based optimization

The gradient approach is used to enhance the population-based technique known as gradient search based optimization (GBO). In GBO, Newton's algorithm determines the search direction. Two primary operators and a collection of vectors are modified to explore the search space more effectively. Only the worst-positioned agents are arbitrarily changed by Eq. (10), as indicated in the research on EHO. This variation mechanism in such methods results in adequate exploitation capability but also leads to sluggish convergence. Additionally, the best-positioned agents are also altered (Eq. 8). However, this phenomenon could decrease population diversity and become redundant once the population settles into a local optimum.

Furthermore, EHO's exploitation potential is only moderately strong, increasing the probability of encountering a local optimum (*Khalilpourazari et al., 2021*). Integrating with GBO may lead to an enhanced solution since it allows for guidance of the search direction during iteration, preventing the possibility of being stuck in a local optimum. The local escape operator (LEO) in GBO can enhance population diversity and prevent overly long periods of stagnation. The suggested method can fully utilize the gradient data in this scenario, thereby enhancing the program's search efficiency (*Hassan et al., 2021*).

## Gradient search rule

To regulate the vector search's direction, Newton's technique yielded the gradient search rule (GSR). A few vectors are included to maintain equilibrium among exploration and exploitation throughout the iterations and speed up convergence:

$$\rho_1 = 2\,rand \times \propto - \propto \tag{20}$$

$$\propto = \left| \beta \times sin\left[ \frac{3\pi}{2} + sin(\beta \times \frac{3\pi}{2}) \right] \right| \tag{21}$$

$$\beta = \beta_{min} + (\beta_{max} - \beta_{min}) \times \left[ 1 - \left( \frac{m}{M} \right)^3 \right]^2 \tag{22}$$

when given values of 1.2 and 0.2, the variables m and M represent the current and maximum number of iterations, respectively. The term "rand" refers to a random number chosen

from the range of 0 to 1. The value of $\alpha$ may be used to regulate the pace of convergence and changes throughout the iterations. Because the value of $\alpha$ is large early in the iteration, the approach may quickly converge on the area where it expects to uncover the optimal answer and improve diversity. The value falls as the loop progresses. As a result, the program can more effectively utilize the studied regions. Based on this, the GSR expression is as follows:

$$GSR = rand \times \rho_1 \times \frac{2\Delta x \times x_n}{(x_{worst} - x_{best} + \varepsilon)} \tag{23}$$

where and signify places of the best and worst agents, and $\varepsilon$ is a small number in the range of $[0, 0.1]$. The suggested GSR's capacity for an arbitrary search improves GBO's capacity for exploration and its capacity to depart from the local optimum, shown below:

$$\Delta x = rand(1:N) \times |step| \tag{24}$$

$$step = \frac{(x_{best} - x_{r1}^m) + \delta}{2} \tag{25}$$

$$\delta = 2 \times rand \times \left( \left| \frac{x_{r1}^m + x_{r2}^m + x_{r3}^m + x_{r4}^m}{4} - x_n^m \right| \right). \tag{26}$$

Let N be a set of random numbers chosen from the interval $[0,1]$. The variable "step" represents the size of each step. The term "global optimal agent" refers to the agent that achieves the best possible outcome on a global scale. It specifically symbolizes the mth dimension of the nth agent. r1, r2, r3, r4 are various integers arbitrarily selected from $[1, N]$.

For a local search, a motion attribute called DM is also set in order to enhance the exploitation capability.

$$DM = rand \times \rho2 \times (x_{best} - x_n) \tag{27}$$

*rand* signifies a random number among $[0,1]$, and $\rho2$ is the step size attribute shown below:

$$\rho2 = 2rrand \times \propto - \propto \tag{28}$$

Finally, the present location of the search agent ($x_n^m$) improved by GSR and DM shown as follow:

$$X1_n^m = x_n^m - GSR + DM. \tag{29}$$

The following is another way of expressing Eq. (29) into the context of Eqs. 14 and 18:

$$X1_n^m = x_n^m - rand \times \rho1 \times \frac{2rx \times x_n^m}{(yp_n^m - yq_n^m + \varepsilon)} + rand \times \rho1 \times (wx_{best} - x_n^m) \tag{30}$$

where $yp_n^m = y_n^m + \Delta x$, $yq_n^m = y_n^m - \Delta x$, and $y_n^m$ is a new created variable defined by the average of $x_n^m$ and $z_{n+1}^m$. Based on Newton's technique, $z_{n+1}^m$ is expressed by:

$$z_{n+1}^m = x_n^m - randn \times \frac{2rx \times x_n^m}{(x_{worst} - x_{best} + \varepsilon)} \tag{31}$$

where $\Delta x$ is definite by Eq. (15), and $x_{worst}$ and $x_{best}$ signify the present worst and best agents, individually. After substituting the present vector $x_n^m$ in Eq. (21) with $x_{best}$, a new vector $X2_n^m$ :

$$X2_n^m = x_{best} - rand \times p1 \times \frac{2\Delta x \times x_n^m}{(yp_n^m - yq_n^m + \varepsilon)} + rand \times p2 \times (x_{r1}^m - x_{r2}^m). \tag{32}$$

According to Eqs. (21) and (23), the new solution $x_n^{m+1}$ can be denoted as:

$$x_n^{m+1} = r_a \times \left[ r_b \times X1_n^m + (1 - r_b) \times X2_n^m \right] + (12r_a) \times X3_n^m \tag{33}$$

$$X3_n^m = x_n^m - \rho1 \times \left( X2_n^m - X1_n^m \right) \tag{34}$$

where $r_a$ and $r_b$ are random numbers among [0,1].

## An LEO is a local egress operator

The method is tuned using the LEO, which increases the probability of obtaining the ideal solution by allowing the program to move away from local optima.

The LEO introduces a solution $X_{LEO}^m$ that performs better and is expressed as:

$$if rand < pr X_{LEO}^m = \begin{cases} X_n^{m+1} + f_1 \times (u_1 \times x_{best} - u_2 \times x_k^m) + f_2 \times p_1 \times \frac{\left[ u_3 \times (X2_n^m - X1_n^m) + u_2 \times (x_{r1}^m - x_{r2}^m) \right]}{2} & rand < 0.5 \\ x_{best} + f_1 \times (u_1 \times x_{best} - u_2 \times x_k^m) + f_2 \times p_1 \times \frac{\left[ u_3 \times (X2_n^m - X1_n^m) + u_2 \times (x_{r1}^m - x_{r2}^m) \right]}{2} & otherwise \end{cases} \tag{35}$$

end

$p_r$ is a predetermined threshold, here $p_r = predef_1$ and $f_2$ is a random number that conforms to the usual normal distribution and is a random number among $[-1,1]$. $u_1, u_2, u_3$ are respectively represented by:

$$u_1 = L_1 \times 2 \times rand + (1 - L_1) \tag{36}$$

$$u_2 = L_1 \times rand + (1 - L_1) \tag{37}$$

$$u_3 = L_1 \times rand + (1 - L_1) \tag{38}$$

where $\mu_1$ is a random integer between [0,1] and $L_1$ is a binary variable with values of 0 or 1 where $\mu_1 < 0.5, L_1 = 1$; otherwise, $L_1 = the$

In conclusion, the resulting solution $x_k^m$ is stated as:

$$x_k^m = L_2 \times x_p^m + (1L_2) \times x_{rand}. \tag{39}$$

A randomly selected population solution is represented by $x_p^m$, $p \in [1, 2, raN]. L_2$, $\mu_2$ is a random number between [0,1] and is a binary variable with values of 0 or 1.

When $\mu_2 < 0.5, L_2 = .5$, otherwise, $L_2 = 0$. xrand is the anew created solution.

$$x_{rand} = X_{min} + rand \times (X_{max} - X_{min}). \tag{40}$$

Finally, the best place of the clan is updated $N^*_{[u_\phi]}$ eliminating the $N'_{[u_\phi]}$. Given is the ideal feature set,

$$O = \{[\kappa_1], [, \kappa_2], \ldots, [\kappa_{tt}]\} \, or \, [\kappa_x], x = 1, 2, \ldots, tt. \tag{41}$$

The dataset is $tt^{th}$ represented after the best feature set has been chosen and $O$ shows the feature set that was chosen.

The pseudocode for UDEHOA is given as follows,

---

**Input:** Feature set $\{[u_1], [u_2], \ldots, [u_d]\}$
**Output:** selected feature sets
**Begin**
**Initialize** $\{[u_1], [u_2], \ldots, [u_d]\}$, population size, Maximum iteration $R_{\max}$
**Set** $R = 1$
**While** $(R \le R_{\max})$**do**
**Compute** elephant's Fitness
**Determine** clan updating operator $N^{R+1}_{[u_\phi], \omega}$ with $\gamma = \frac{1}{N^R_{[u_\phi]+1} - N^R_{[u_\phi]}}$

**Determine** clan separating operator $N'_{[u_\phi]}$

**Evaluate** fitness of $N^{R+1}_{[u_\phi], \omega}$

**If fitness** of $N^{R+1}_{[u_\phi], \omega}$ higher **Then**

**Update** clan position $N^*_{[u_\phi]}$

**for** n =1:N do
**for** i =1: dim do
Arbitrarily selects r1,r2,r3,r4 in the range of [1,N]
Estimate GSR and DM based on (14) and (18)
Calculate , ,
Calculate
**End for**
**If** rand<pr then
Generate
**End if** Calculate and update the fitness according to each position
**End for**

**Else**
$R = R + 1$
**End If**
**End While**
**Return** optimal feature set $N^*_{[u_\phi]}$
**End**

---

## Decision making

Fuzzy logic fuzzifies crisp inputs, develops decision-making rules, and fuzzes crisp feature values when the associated feature sets $\{[\kappa_x]\}$ are selected.

Initially, in the fuzzy logic, fuzzify the input feature set using the membership function. Here, to fuzzify $\{[\kappa_x]\}$ trapezoidal membership function is used, which is denoted as,

$$\nabla([\kappa_x], w, xx, Dia-cls, Hea-cls) = \max\left(\min\left(\frac{[\kappa_x]-w}{xx-w}, 1, \frac{Hea-cls-[\kappa_x]}{Hea-cls-Dia-cls}\right), 0\right)$$

(42)

where, $\nabla()$ depicts the trapezoidal membership function. $w, xx, dia-cls, z$ heart and diabetes feature values, class, and input parameters. Rule-based decision-making follows,

$$normal = \{1 \text{ if } dia-cls = 0 \ \& \ Hea-cls = 0$$

(43)

$$Dia-risk = \begin{cases} 2 \text{ if } dia-cls = 1 \& Hea-cls = 0 \& \alpha(dia) > xx \\ 3 \text{ if } dia-cls = 1 \& Hea-cls = 0 \& \alpha(0) < xx < \alpha(dia) \end{cases}$$

(44)

$$Hea-risk = \begin{cases} 5 \text{ if } dia-cls = 0 \& Hea-cls = 1 \& \alpha(Hea) > w \\ 6 \text{ if } dia-cls = 0 \& Hea-cls = 1 \& \alpha(0) < w < \alpha(Hea). \end{cases}$$

(45)

The normal patient, low risk of diabetes, high risk of diabetes, low risk of heart disease, and high risk of heart disease have decision rules 1, 2, 3, 4, 5, and 6. $Dia-risk, Hea-risk$ depicts the diabetic and heart risk respectively, $Dia-cls, Hea-cls$ depicts the diabetic and heart-disease classes. The data aggregation means the value of heart disease and diabetes is given as $\alpha(Hea)$ and $\alpha(Dia)$ respectively. Similarly, also established were patients' low and high heat and diabetes risk determination guidelines. The diabetic patient has high risk of stroke. Finally, defuzzification gives the feature's sharp value. For all the feature sets, the crisp values are given as,

$$V = \{g_1, g_2, \ldots, g_{cv}\} \text{ or } g_{\bar{\lambda}}, \bar{\lambda} = 1, 2, \ldots, cv.$$

(46)

Here, $g_{cv}$ is crisp value of $[\kappa_{tt}]$ after applying fuzzy rules, and $V$ depicts the defuzzified feature values. The AWBi-LSTM classifier receives fuzzy logic feature values, which predicts and classifies heart disease and diabetes.

## Adaptive weight bi-directional long short-term memory for classification and risk prediction

One aspect that sets a recurrent neural network (RNN) apart from a feed-forward network is that the neurons in hidden layers receive feedback, incorporating both prior and current states. Theoretically, RNNs can learn features from time series of any length. However, experiments indicate that RNN performance may be limited due to vanishing gradient or gradient explosion. To address these gradient problems, the LSTM network was developed, introducing a core element known as the memory unit.

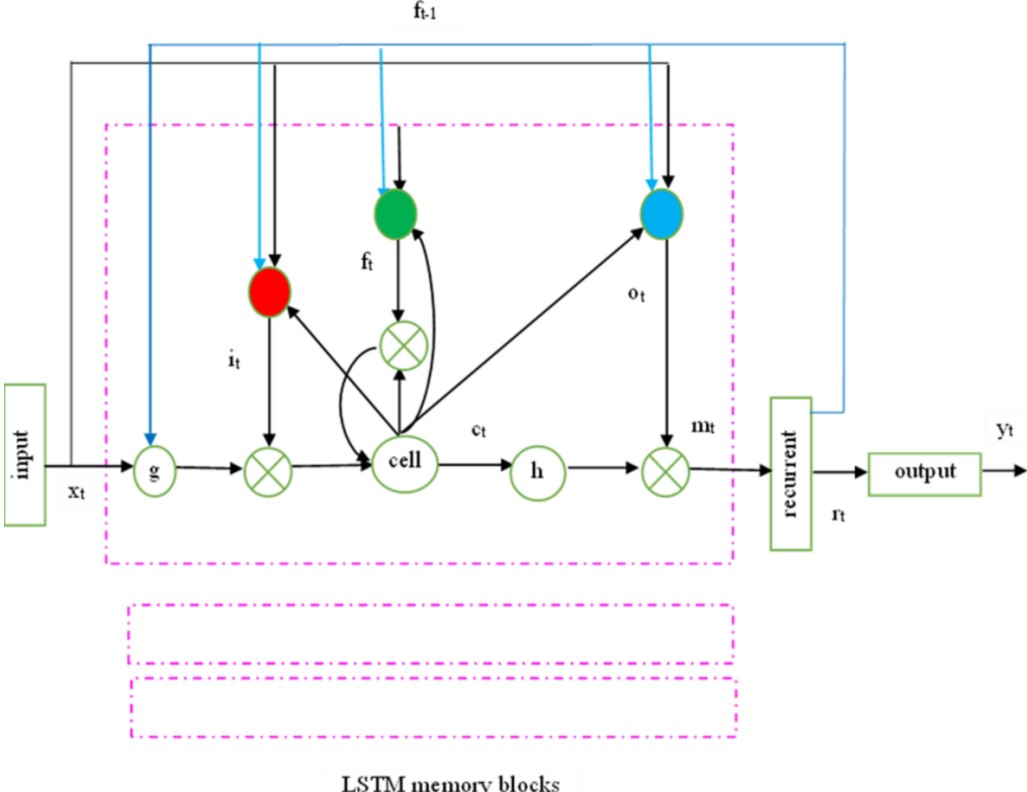

**Figure 2** **LSTMP RNN architecture.**

The recurrent hidden layer of the LSTM has specialized memory blocks as shown in Fig. 2. These memory blocks include self-connected memory cells that store the network's temporal state and gates for information flow control. Each memory block in the actual model includes input and output gates. While the output gate controls the output flow associations of the cell activations into the remainder of the network, the input gate controls the flow of input activations into the memory cell. To decide how much memory cell should be decimated from the present memory cell, the forget gate was subsequently added to the memory block. This method fixes LSTM models' inability to interpret continuous input streams without subsequences. To adaptively forget or reset the cell's memory, the forget gate adjusts its internal state before giving it back as input.

Keyhole connections from internal cells to gates in the same cell enable the contemporary LSTM to learn output timing. The final gate, 'o', called after the output gate, controls the memory unit's output activation information and flows into the network.

The following equations are used iteratively from $t = 1$ to T to estimate the network unit activations and convert an input sequence x $= (x_1,.., x_T)$ to an output sequence y $= (y_1,.., y_T)$. In the LSTM, W terms indicate weight matrices (*e.g.*, $W_{ix}$), diagonal weight matrices ($W_{ic}$, $W_{fc}$, $W_{oc}$), bias vectors (bi), logistic sigmoid function ($\sigma$), and input, forget, output, and cell activation vectors (i, f, o, c), the cell input and output activation

functions are g and h, respectively, while the network output activation function is Softback.

In the LSTM classifier, weights can be considered as the connection strength. Weight is accountable for the degree of effect that will be put on the output when a modification in the input is seen. A lesser weight value will not change the input, whereas a larger weight value will significantly modify the output. Every component includes weights corresponding to all its inputs from the earlier layer, in addition to the input from the earlier time step. Associative memory applying fast weights is a short-term memory technique, which considerably enhances the memory capability and time scale of RNNs.

Bi-LSTM extends LSTM; it is helpful in discovering the associations between datasets. The same output layer is connected to two LSTM networks, one of which is forward-looking and the other of which is backward-looking. The same output layer is connected to two LSTM networks, one of which is forward-looking and the other of which is backward-looking. to select the features optimally. In this research work, Rand Index (RI) is regarded as the fitness function for optimally selecting the features from the dataset. The same sequence of data is used for training both. Three gates exist, which are known as input, forget, and output gate, in an LSTM unit. These gates operate on the basis expressions (Eqs. (47)–(52)),

$$i_t = \sigma \left( w_i [h_{t-1}, x_t] + b_i \right) \tag{47}$$

$$f_t = (w_f [h_t, x_t] + b_f) \tag{48}$$

$$o_t = 4 (w_o [h_{t48}, x_t] + b_o) \tag{49}$$

$$\tilde{c}_t = \tanh (w_c [h_{tta}, x_t] + b_c) \tag{50}$$

$$c_t = f_t * c_{t50} + i_t * \tilde{c}_t \tag{51}$$

$$h_t = o_t * \tanh(c_t) \tag{52}$$

Here, $w_i$, $w_f$, and $w_o$ refer to the weights of LSTM, and $b_i$, $b_f$, and $b_o$ indicate the biases. $i_t$ stands for the input gate, $f_t$ signifies the forget gate, and $o_t$ represents the output gate. $x_t$ signifies the input vector and $h_t$ stands for the output vector. $c_t$ refers to the cell state and t $\tilde{c}_t$ implies the candidate of the cell state. In the case of the forward LSTM, expressed as $\vec{h}_t \rightarrow$ liex$_t$, $\vec{h}_t$ . In accordance, the backward LSTM is with $\vec{h}_t$ith acx$_t$, $\vec{h}_{tt}$. Both $\vec{h}$ and $\vec{h}$ constitute the output of Bi-LSTM at a time,

$$h_t = \left[ \vec{h}_t; \vec{h}_t \right] \tag{53}$$

Especially, the optimization of the Bi-LSTM (*i.e*, weight values) is performed dynamically. Therefore, the fitness function is modifiable and may evaluate the fitness score of each Bi-LSTM model throughout the weight formation process in their different training processes. This suggests that the fitness ratings evaluated in different generations are not comparable. In the AWBi-LSTM algorithm, the mutation parameter is used for generating new weights according to the mean value of a feature. The selection technique of AWBi-LSTM is denoted as $\{BiTMiM_i\}_{i=1}^{\lambda}$, and it is ranked based on their fitness function $F_i$, the highest mean weight values ($\mu$) are chosen as the top feature. The pseducode for AWBi-LSTM is as follows:

---

**Adaptive Weight Bi-Directional Long Short-Term Memory (AWBi-LSTM)**

**Input:** Total quantity of samples in the dataset N, the quantity of mutations $n_m$, the batch size m, dataset D, and initial weight $w_0$,

**Output:** Best chosen features from the dataset

Start $w = w_0$

Initialize model parameter $w_0$

**for** $i = 1$ to $m/(Nn_m)$

param ⟵w save model parameters

**for** $j = 1$ to N

**for** $k = 1$ to $n_m$

M(param) allocate parameters to the system

obtain a set D as input $x_i$ of AWBi-LSTM;

**switch(k)**

case1: $loss_{square}$, $param_{square}$ ⟵M($x_i$, square, param)

case2: $loss_{abs}$, $param_{abs}$ ⟵M($x_i$, abs, param)

case3: $loss_{huber}$, $param_{huber}$ ⟵M($x_i$, huber, param)

**end switch**

if $k = 1$ to $n_m$

$loss_{min}$ ⟵ min($loss_{square}$, $loss_{abs}$, $loss_{huber}$)

$param_{new}$ ⟵($loss_{min}$, $param_{square}$, $param_{abs}$, $param_{huber}$)

w ⟵$param_{new}$

**end for**

**end for**

**End**

---

## Data security using MD5-ECC

After predicting the risk level, the database securely stores the data. For secure storage, the MD5-ECC technique is utilized. Elliptic curve cryptography (ECC) algorithm, despite requiring very little computation and a very small key size compared to other techniques, is secure. However, the complexity and difficulty of this algorithm increase the probability of implementation errors and reduce the system's security. Therefore, MD5-ECC is suggested to enhance the security level of ECC. In ECC, only the public and private types of keys are produced; however, MD5-ECC adds a third type of key known as the secret key by using the MD5 hash function to increase the framework's safety. The use of MD5 is intended to increase the complexity of ECC. As attackers attempt to assault the data, the complexity of the algorithms rises. The produced secret key is used for decryption as well as encryption. Thus, the MD5-ECC, whose mathematical description is shown here,

$$m^2 = x^3 + ax + b \tag{54}$$

here, a and b mean the integers. In the suggested study, '3' different types of keys must be established.

Step 1: regarded point $B_p$ as the curve's base point. Create the public key A using Eq. (20).

$$A = (K * B_p) \tag{55}$$

here, according to K, the chosen private key falls between (1 and n − 1).

Step 2: By adding the salt value to the public key and using the MD5 hash method to get a hash value from this value, you may establish a unique secret key. The novel key $S_k$ is developed as

$$S_k = MD5(A \| S_d) \tag{56}$$

here $S_d$ indicates a salt value that is randomly preferred.

Step 3: Encrypt data using the secret key and public key, a curve point. The secret key is combined with the encryption algorithm in the suggested MD5-ECC. The encrypted data consists of '2' ciphertexts, which are mathematically denoted as,

$$E_1 = (R * B_p) + S_k \tag{57}$$

$$E_2 = D + (R * A) + S_k \tag{58}$$

E1 and E2 represent encrypted text 1 and 2, respectively, whereas R represents a random integer, and D represents data. Since the decryption, the original data has been obtained.

Step 4: By performing the encryption's reverse operation, the data can be decrypted. Decryption subtracts the secret key from the equation, represented as

$$D = ((C_2 - K) * C_1) - N_k. \tag{59}$$

Hence, with the ECC cryptography technique, he medical database properly stores healthcare outcomes.

# RESULTS AND DISCUSSION

This section presents an evaluation of the performance of the proposed approach. The proposed approach is implemented using MATLAB. Performance analysis and comparative analysis were conducted to demonstrate the effectiveness of the work. The Stroke Prediction dataset, accessible at https://www.kaggle.com/datasets/fedesoriano/stroke-prediction-dataset, is used to evaluate the efficacy of the suggested risk prediction model.

This dataset determines the likelihood of a person having a stroke based on 11 input characteristics, including age, gender, marital status, hypertension, profession, glucose level, BMI, blood pressure, existing diseases, chest discomfort, and smoking status. The dataset comprises more than 5,000 samples. Here is the link to the Kaggle Stroke Prediction Dataset.

CSV dataset: The heart stroke prediction dataset from Kaggle was utilized as a CSV dataset. The dataset contains 11 parameters such as age, ID, gender, work type, hypertension, residence type, average level of glucose, heart disease, body mass index (BMI), smoking behavior, marital status, and stroke.

Description of the dataset: There are 11 attributes in the dataset, and each one identifies whether the data is categorized or numerical.

ID: This component displays a person's distinctive identifier. Information that can be computed.

Age: This trait serves as a proxy for the person's age. details on classifications

Gender: This attribute reveals the person's gender. Information that is obtainable.

Hypertension: This characteristic shows if the person has high blood pressure or not. details regarding the classifications.

Work type: This characteristic describes a person's employment. details regarding the classifications.

Residence type: This characteristic reflects the person's current situation.

Heart disease: This characteristic suggests that the person may have heart disease. Information that is calculable.

Average glucose level: This statistic shows the average level of a person's blood sugar. Information that is calculable.

BMI: The acronym BMI stands for ''numerical data.'' The BMI (body mass index) of a person is referred to in this attribute.

Ever married: details from the group. This attribute denotes a person's marital status.

Smoking status: Statistical data broken down by category. This trait reveals a person's smoking status.

Stroke: This characteristic indicates whether someone has suffered a stroke. The complete attribute dash represents the target class, while the remaining attributes represent the feature classes.

A training dataset, which makes up 80% of the total, is separated from a testing dataset in the input dataset. The term ''training dataset'' refers to the collection of data that a

machine learning model learns from. Testing datasets are used to show how successful the trained model.

## Evaluation metrics

The proposed research method is applied on this dataset and performance evaluation is made of metrics such as precision, accuracy, error rate, recall, F-measure, and number of rules. These metrics are evaluated based on correct and wrong prediction parameters with True Positive (TP), False Positive (FP), True Negative (TN), and False Negative (FN).

The performance evaluation in the graphical representation is defined in the following.

**Precision:** The accuracy of a forecast is determined by the number of predicted positive observations. A small rate of false positives indicates great precision. The formula below explains it (3.10),

$$Precision = TP/TP + FP. \tag{60}$$

Precision is used to calculate the ratio among the correctly predicted stroke data and the stroke prediction data given in database.

**Recall:** The percentage of accurately anticipated positive results to all class findings is recalled.

$$Recall = TP/TP + FN \tag{61}$$

It computes the ratio of successfully predicted stroke data to database-positive stroke data.

**Sensitivity:** Sensitivity is the fraction of relevant data that were regained.

$$Sensitivity = \frac{TP}{(TP + FN)} \tag{62}$$

**Specificity:** Specificity refers to the probability of negative matches, conditioned on truly being negative.

$$Specificity = \frac{TN}{(TN + FP)} \tag{63}$$

**F-Measure:** The precision and recall weighted average is termed the F-Measure. This means that erroneous positives and negatives are reflected in this score.

$$F\text{-}measure = 2 * (Recall * Precision)/(Recall + Precision) \tag{64}$$

**Accuracy:** Since accuracy is the logical efficiency metric, it can be defined as the ratio of properly predicted data to all data.

$$Accuracy = (TP + TN)/(TP + FP + FN + TN) \tag{65}$$

The ratio between the entire amount of stroke data and the accurately anticipated stroke data is measured by this statistic. Here, the correctly predicted stroke data would be divided with total number of stroke data for measuring the accuracy value.

**Negative predictive value (NPV):** The negative predictive value is definite as follows,

$$NPV = \frac{TP}{(TN + FN)} \tag{66}$$

**Table 1  Comparative analysis of the proposed HKGA model.**

| Techniques | Clustering time (sec) |
|---|---|
| Proposed HKGA | 1.181 |
| DH-CC-KC | 1.239 |
| K-means | 1.293 |
| GMM | 2.636 |
| KNN | 5.174 |

**Matthews correlation coefficient:** This method of determining the Pearson product-moment correlation coefficient among actual and predicted data utilizes a contingency matrix. Regarding M's entries, MCC is:

$$MCC = \frac{TP.TN - FP.FN}{\sqrt{(TP+FP).(TP+TN).(TN+FP).(TN+FN)}} \tag{67}$$

## Performance analysis of HKGA

The suggestedHKGAperformance is analyzed with the existing methods, such as K-means, Gaussian Mixture Model (GMM) algorithm, and K-Nearest Neighbor (KNN) based on the time consumed for clustering.

The clustering times of the suggested and current techniques are illustrated in Table 1. The proposed HKGA method takes a clustering time of 1.181 s. However, existing methods consumed a significantly longer time for clustering. The partial derivative of the Hamiltonian in conventional K-means showed improvement in clustering time. The above analysis indicates that the proposed method requires less time for clustering than existing methods.

## Performance analysis of ICA

The suggested ICA performance is compared with the prevailing technique like SS-PCA, PCA, Linear Discriminative Analysis (LDA), and Gaussian Discriminative Analysis (GDA) based on the metrics, such as Peak Signal-to-Noise Ratio (PSNR), Mean Square Error (MSE) and R-Square.

The performance of the suggested ICA along with present approaches is assessed in Table 2 regarding the quality metrics, PSNR and MSE. Higher PSNR and lower MSE values indicate the superior performance of the feature reduction strategy. Compared to the current SS-PCA, PCA, LDA, and GDA techniques, the PSNR value obtained by the suggested approach is 2.7 dB higher. With a low error value of 0.01010, the suggested ICA outperforms the traditional frameworks. The current PCA performs better now that the shell sorting procedure has been used. As a result, the recommended ICA effectively reduces the characteristics.

Table 2's performance numbers can only be improved when R-Square, a statistical metric, is high. The R-Square values of the current models were 0.756 (SS-PCA), 0.653 (PCA), 0.374 (LDA), and 0.175 (GDA), whereas the recommended approach had an R-Square value of 0.810. Thus, the analyses show that the suggested strategy is significantly superior to others.

**Table 2** Performance analysis of the proposed ICA.

| Techniques/Metrics | PSNR (dB) | MSE | R-Square |
|---|---|---|---|
| ICA | 40.99 | 0.01010 | 0.810 |
| SS-PCA | 39.87 | 0.01015 | 0.756 |
| PCA | 38.85 | 0.01141 | 0.653 |
| LDA | 37.94 | 0.01268 | 0.374 |
| GDA | 31.25 | 0.02738 | 0.175 |

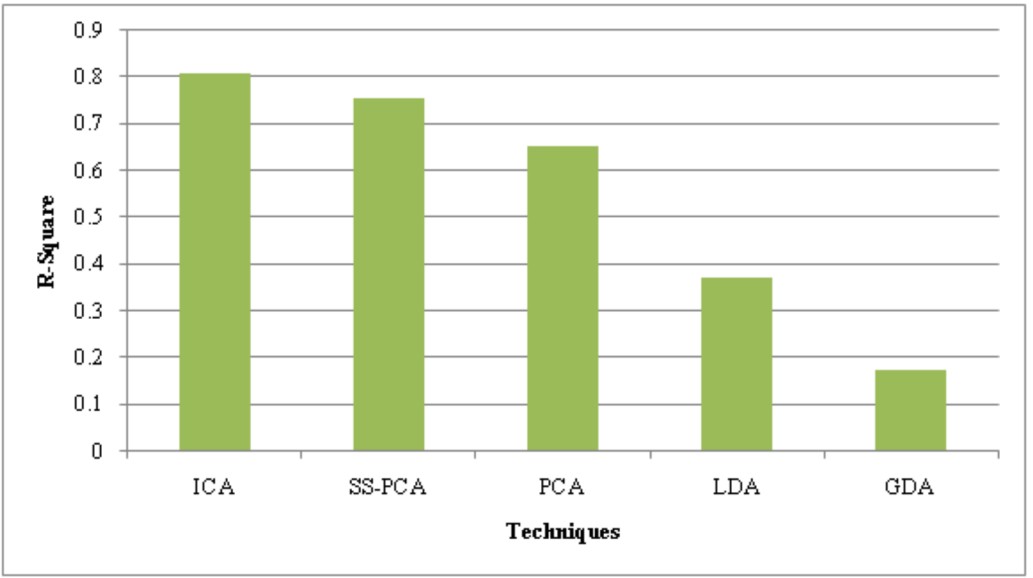

**Figure 3** Performance analysis of the proposed AWBi-LSTM classifier method.

## Performance analysis of AWBi-LSTM classifier

The performance of the proposed AWBi-LSTM classification model is examined and compared with other existing techniques like RBM, Convolutional Neural Networks (CNN), Deep Neural Networks (DNN), and Recurrent Neural Networks (RNN). Based on quality metrics like F-measure, False Positive Rate (FPR), False Recognition Rate (FRR), False Negative Rate (FNR), Net Present Value (NPV), MCC, and confusion matrix, the suggested technique is compared with the current methods.

In Fig. 3, the performance of the proposed and existing approaches are analyzed according to Sensitivity, Specificity, accuracy, and precision values shown in Table 3. The proposed method achieved a sensitivity of 98.81%, whereas the existing PLD-SSL-RBM, RBM, CNN, DNN, and RNN have 98.42%, 88.25%, 87.28%, 85.83%, and 85.71%, respectively, Likewise, the suggested technique has sensitivity, specificity, accuracy, and precision of 98.81%, 99.80%, 99.65%, and 98.64%, respectively, which is higher than the existing methods. Semi-Supervised Learning and Power Lognormal Distribution have

**Table 3** Performance analysis of the proposed AWBi-LSTM classifier method.

| Techniques/Metrics | Sensitivity | Specificity | Accuracy | Precision |
|---|---|---|---|---|
| ProposedAWBi-LSTM | **98.81** | **99.80** | **99.65** | **98.64** |
| PLD-SSL-RBM | 98.42 | 99.73 | 99.55 | 98.42 |
| RBM | 88.25 | 99.61 | 99.02 | 97.56 |
| CNN | 87.28 | 99.40 | 98.52 | 97.03 |
| DNN | 85.83 | 99.12 | 98.31 | 96.31 |
| RNN | 85.71 | 99.03 | 98.01 | 96.21 |

**Notes.**
The bold values indicate that the proposed system achieves the highest results for all metrics.

**Table 4** Performance analysis of proposed AWBi-LSTM.

| Techniques/Metrics | F-measure (%) | NPV (%) | MCC (%) |
|---|---|---|---|
| ProposedAWBi-LSTM | 98.89 | 99.84 | 98.52 |
| PLD-SSL-RBM | 98.42 | 99.73 | 98.16 |
| RBM | 88.25 | 98.04 | 86.29 |
| CNN | 87.29 | 97.88 | 85.16 |
| DNN | 85.83 | 97.63 | 83.47 |
| RNN | 85.71 | 97.61 | 83.33 |

enhanced the performance of the classifier to a greater extent. Overall, the performance analysis reveals that the proposed method accurately classified the risk classes.

Table 4 exhibits the performance of the suggested AWBi-LSTM according to F-Measure, MCC, and NPV. The proposed method's F-Measure, NPV, and MCC values are 98.89%, 99.84%, and 98.52%, respectively, whereas the existing methods provide comparatively lower performance. The proposed AWBi-LSTM approach outperformed all other current methods in this performance comparison.

Figure 4 illustrates the analysis of the proposed method alongside existing methods based on F-measure, NPV, and MCC, which are values contributing to false predictions. The proposed model attained higher F-measure, NPV, and MCC values. Hence, it can be concluded that the proposed method achieved better performance and accurately classified the classes.

Figure 5 illustrates the performance of the classification model using the confusion matrix. The confusion matrix is employed to evaluate the model's accuracy by comparing the predicted class with the actual class. The classifier's accuracy is calculated as the percentage of occurrences that are successfully classified. The confusion matrix clearly demonstrates that the proposed AWBi-LSTM provides better accuracy. Thus, the phases are more effectively categorized by the proposed framework.

## Comparative measurement with literature papers

Here, the effectiveness of the proposed approach is compared with traditional approaches like Classification and Regression Tree (CART) (*Carrizosa, Molero-Río & Morales, 2021*), Stacked Sparse Auto-Encoder and Particle Swarm Optimization (SSAE-PSO) (*Mienye &*

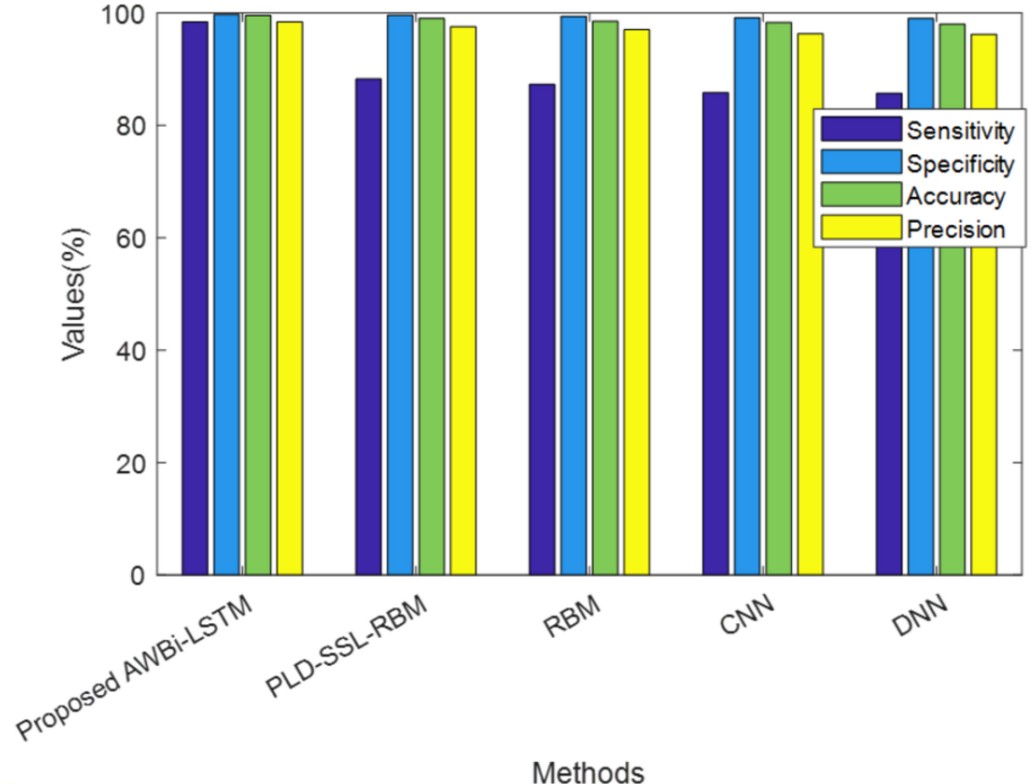

**Figure 4** Comparative analysis of the proposed AWBi-LSTM and the existing methods.

**Table 5** Comparative analysis of the proposed model and previous studies.

| Techniques/Metrics | Accuracy (%) | Precision (%) | F-Measure (%) |
|---|---|---|---|
| CART | 91 | 92 | 91 |
| SSAE- PSO | 97.3 | 94.8 | 97.3 |
| SA | 90.24 | 92 | 90 |
| PLD-SSL-RBM | 99.55 | 98.42 | 98.42 |
| Proposed AWBi-LSTM | 99.65 | 98.64 | 98.89 |

*Sun, 2021*), and Stacking Algorithm (SA) (*Abdollahi & Nouri-Moghaddam, 2022*) based on precision, accuracy and F-Measure obtained using the Framingham dataset.

Table 5 presents a comparative analysis of the AWBi-LSTM model suggested in this study with the models used in previous research. The analysis reveals that the suggested framework was more efficient than other frameworks in predicting diabetes and heart disease.

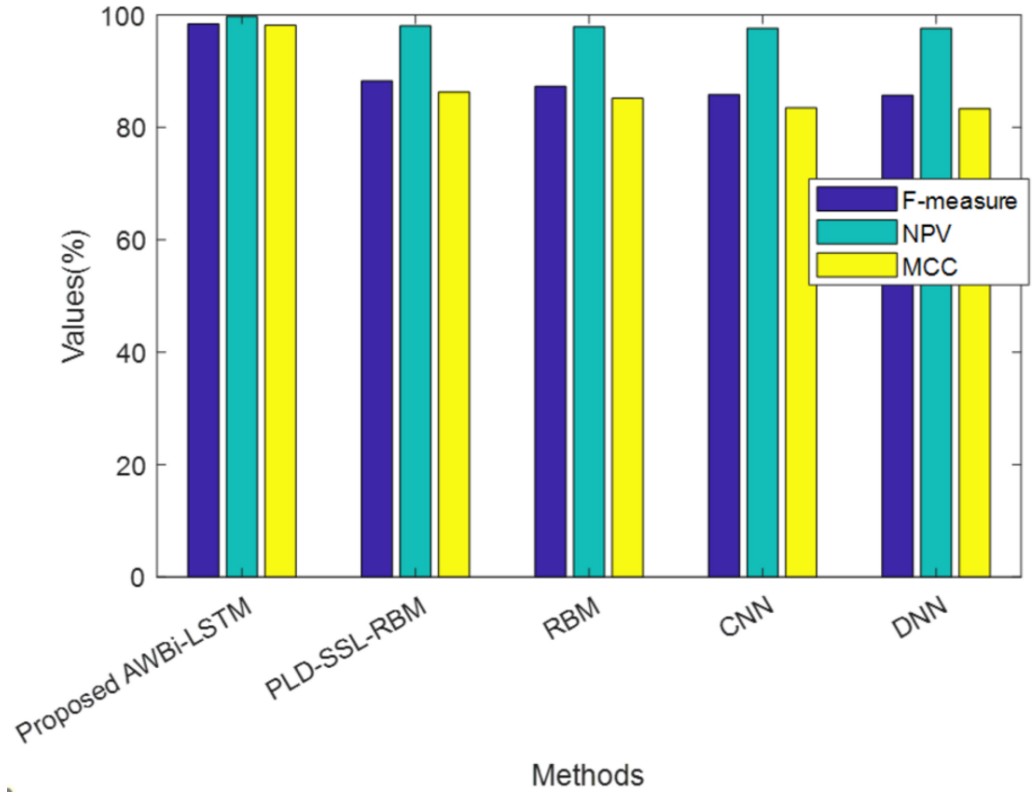

**Figure 5** Confusion matrix for the proposed AWBi-LSTM.

## CONCLUSION

This work proposes a novel framework termed AWBi-LSTM-based stroke disease prediction model for IoT. Pre-processing, feature reduction, feature correlation, decision-making, optimum feature set selection, classification, risk prediction, and encryption are the stages through which the framework functions. Subsequently, performance assessment uses the stroke prediction dataset to compare the proposed technique to existing systems. From the experimental analysis, the proposed framework achieves an accuracy of 99.65%, precision of 98.64%, and F-measure of 98.89%. The proposed approach required 1 s less clustering time than the current system. Thus, the proposed approach proves to be better and more efficient than other present techniques. However, the proposed work, which focuses solely on diabetes and heart stroke risk analysis, provides better results. The future scope of this work is to apply this proposed algorithm to a larger dataset and explore other deep learning models. The use of larger datasets could significantly enhance the reliability of the model. The deployment of deep learning architectures would help determine the likelihood of a stroke in adult patients. Early identification allows individuals to promptly obtain stroke treatment and then reconstruct

their lives after the incident. Hence, future research endeavors to enhance the precision of early identification while minimizing the occurrence of errors.

### Funding

The authors received no funding for this work.

### Competing Interests

The authors declare there are no competing interests.

### Author Contributions

- S Thumilvannan conceived and designed the experiments, performed the experiments, analyzed the data, performed the computation work, prepared figures and/or tables, authored or reviewed drafts of the article, and approved the final draft.
- R Balamanigandan performed the experiments, authored or reviewed drafts of the article, and approved the final draft.

### Data Availability

The Stroke Prediction dataset is available at Kaggle: https://www.kaggle.com/datasets/fedesoriano/stroke-prediction-dataset.

### Supplemental Information

Supplemental information for this article can be found online at http://dx.doi.org/10.7717/peerj-cs.2196#supplemental-information.

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
