# Peer review of "A novel adaptive weight bi-directional long short-term memory (AWBi-LSTM) classifier model for heart stroke risk level prediction in IoT"

_PeerJ Computer Science, doi:10.7717/peerj-cs.2196_

## Round 0.1 · original submission · Minor Revisions

This work tackles an important topic. The authors need to satisfy all the comments from the reviewers. Also, the abstract needs to be shorter, more informative, and focused on the contribution without using any titles or bolded words. The paper needs more writing revision. The comparison of the proposed contribution with existing methods needs more discussion.

**Language Note:** The review process has identified that the English language must be improved. PeerJ can provide language editing services - please contact us at [email protected] for pricing (be sure to provide your manuscript number and title). Alternatively, you should make your own arrangements to improve the language quality and provide details in your response letter. – PeerJ Staff

Reviewer 1 ·

Basic reporting

The paper presents an innovative approach to a novel framework termed AWBi-LSTM -based diabetes and stroke disease prediction models for IoT. It employs methods like Hybrid Genetic with Kmeans Algorithm (HKGA) for data-preprocessing and Independent Component Analysis (ICA) for feature reduction, T-test-based Uniform Distribution- gradient search rule based elephant herding optimization for cluster analysis (UD-GSRBEHO) (T-test-UD- GSRBEHO) for feature correlation, along with Adaptive Weight Bi-Directional Long Short-Term Memory (AWBi-LSTM) for classification and risk prediction . The paper's strengths include its novel approach to disease prediction models for IoT, clear methodological framework, and potential applicability in diverse fields. However, the paper could be improved in the following aspects:
1)Introduction and Literature Review: The introduction effectively sets the context for the research, but it could benefit from a more detailed discussion of previous work in this area. Including a broader range of literature would strengthen the argument for the necessity of the proposed method.Kindly review the formatting throughout the manuscript, including text style, size, formulas, tables, figures, etc.Minor revisions are needed to improve the style of the article, as it contains spelling and punctuation errors.
2) Conclusion and Future Work: The conclusion succinctly summarizes the findings but could be expanded to include recommendations for future research based on the observed results and limitations.

Experimental design

1)Methodology: The use of HKGA , ICA ,T-test-UD- GSRBEHO and AWBi-LSTM is well-explained, but the paper could provide more justification for choosing these methods over others. Additionally, details on the dataset used, including its size and diversity, would help in assessing the robustness of the methodology.
2)Data Analysis and Results: While the results show promise, the accuracy rate of 99.65% indicates potential areas for improvement. The paper should discuss possible reasons for this level of accuracy and propose ways to enhance it.
3)Discussion: The discussion section is crucial for interpreting the results. The paper could expand on how the findings compare with existing theories and practices in the field. Discussing the implications of the research and potential applications would add depth to the paper.

Validity of the findings

no comment

·

Basic reporting

1. There are couple of places where sentence formulation can be better. ex. Line 16-18 (Please see attached reviewed PDF)
2. Character space is required in multiple places, Line 29, 522, 527, 528, etc. (Please see highlighted text in attached PDF).
3. Section Algorithm 3.1 needs proper indentation for the readers to understand it correctly. (Please see comments in attached PDF)

Experimental design

The scope of work is well defined. Literature review backs the suggested technique.
The suggested method is well defined. Algorithm needs appropriate indentation.

Validity of the findings

The data used has 5000 samples. It would be great if we can note how large was the final dataset after cleaning or after removing missing data. The result and discussion is well written with attention to detail by comparing the method with traditional details.

---

## Round 0.2 · accepted · Accept

Dear Authors,

Your paper has been accepted for publication in PEERJ Computer Science.
Thank you for your fine contribution.

·

Basic reporting

Basic reporting is appropriate.

Experimental design

Design and methods are appropriate.

Validity of the findings

Outcomes are appropriately defined.